# Quantifying Positive and Negative Human-Modified Droughts in the Anthropocene: Illustration with Two Iranian Catchments

**Elham Kakaei** [1,2], **Hamid Reza Moradi** [1,*], **Alireza Moghaddam Nia** [3] and
**Henny A.J. Van Lanen** [2]

1   Department of Watershed Management Engineering, College of Natural Resources and Marine Sciences, Tarbiat Modares University, Noor 46414-356, Iran; kakaei.elham@gmail.com
2   Hydrology and Quantitative Water Management Group, Wageningen University, 6700 AA Wageningen, The Netherlands; henny.vanlanen@wur.nl
3   Faculty of Natural Resources, University of Tehran, Karaj 1417466191, Iran; a.moghaddamnia@ut.ac.ir
*   Correspondence: hrmoradi@modares.ac.ir; Tel.: +98-1144553101

**Abstract:** In the Anthropocene, hydrological processes and the state of water in different parts of the terrestrial phase of the hydrological cycle can be altered both directly and indirectly due to human interventions and natural phenomena. Adaption and mitigation of future severe droughts need precise insights into the natural and anthropogenic drivers of droughts and understanding how variability in human drivers can alter anthropogenic droughts in positive or negative ways. The aim of the current study was expanding the "observation-modelling" approach to quantify different types of natural and human droughts. In addition, quantifying enhanced or alleviated modified droughts was the second parallel purpose of the research. The main principle of this approach is the simulation of the condition that would have happened in the absence of human interventions. The extended approach was tested in two Iranian catchments with notable human interventions and different climatic conditions. The drought events were identified through hydrological modelling by the Hydrologiska Byråns Vattenbalansavdelning (HBV) model, naturalizing the time series of hydrometeorological data for a period with no significant human interventions, and anomaly analysis. The obtained results have demonstrated that both catchments were almost the same in experiencing longer and more severe negative modified droughts than positive ones because of the negative pressure of human activities on the hydrological system. A large number of natural droughts have also been transformed into modified droughts because of the intensive exploitation of surface and sub-surface water resources and the lack of hydrological system recovery. The results show that the extended approach can detect and quantify different drought types in our human-influenced era.

**Keywords:** anthropogenic drought; human intervention; anomaly analysis; drought management; Iranian catchment

---

## 1. Introduction

All around the world and under global warming, drought has been a major concern of sustainable water resources management because of the vast negative impacts, e.g., on the environment, food security, agriculture, energy, access to safe drinking water, public health, and public hygiene [1–3]. In classical definitions, drought has been defined as a natural phenomenon due to climate variability (single drought driver). However, in those definitions, the influence of interaction between anthropogenic activities and natural processes on drought phenomena, propagation, and characteristics have not been considered [4–7]. The environment, hydrological processes and the state of water in

different parts of the terrestrial phase of the hydrological cycle can be altered both directly and indirectly due to human effects on water resources [8]. In our human-influenced era, human interventions and hydro-climate variability are known as the main factors for the occurrence of hydrological drought [6,9]. It is undeniable that the multi-directional relationship between human influences, as societal processes, and natural drought processes have a key role in drought management [10] in the Anthropocene [6,7]. In order to get a better vision of the interplay between humans, climate and hydrology, hydrologists need to consider drought as a complex interdisciplinary phenomenon instead of a purely natural phenomenon. It should be noted that in areas with high levels of human intervention, using traditional drought indices will lead to poor assessment. Therefore, drought research needs to modify from single drivers and uni-directional to multiple drivers and multi-directional [10]. In order to overcome the gaps in understanding drought phenomena in the Anthropocene, water resources management needs a new framework to involve human processes as human drivers in drought definition.

Recently, there has been a call to introduce new drought definitions and approaches by which the anthropogenic influence on drought phenomenon has been included [11,12]. Consequently, drought has been defined by combining natural and anthropogenic drivers of drought [7]. Climate-induced drought is a type of hydrological drought caused only by natural drought drivers. In this type of drought, the anthropogenic drivers of drought are non-existent. Human-induced drought has been increasing due to human activities and anthropogenic drivers of drought in the absence of natural drivers [6]. Finally, as a combination and simultaneously interaction of both natural and anthropogenic drivers, human-modified drought has been categorized as a third type of drought in the Anthropocene. Long-term human modification has the potential to mitigate or exacerbate human-induced drought severity, so that there can be both positive and negative human-modified drought. In vulnerable arid and semi-arid environments, human-modified drought has become a common natural hazard mainly because of the low annual precipitation, high level of human pressure on the water system, due to agricultural and industrial development, and rapid population growth.

Until now, different approaches have been developed to quantify human influence on hydrological drought, e.g., (i) observation-modelling; (ii) upstream-downstream; (iii) paired catchment; (iv) pre-post-disturbance; and (v) large-scale screening.

The "observation-modelling" approach (i) is based on making a comparison between naturalized simulated and human-influenced observed drought events [11]. This method is based on the availability of hydrological and meteorological data for the natural and disturbed period, irrespective of the region. It is possible to choose different kinds of hydrological models in the framework, e.g., stochastic, lumped and conceptual or physically based models. The kinds of required hydrometeorological variables depend on the choice of hydrological model. The "upstream-downstream" approach (ii) compares upstream drought events, which are not supposed to be affected by disturbance, with drought events downstream. This method needs observation time series of the variable of interest. Existing possible non-linear relationships between upstream and downstream can be a source of uncertainty in human drought studies using this method [11,12]. The "paired catchment" approach (iii) can quantify the human effect through comparing two nearby catchments with similar physical characteristics including the human affected catchment and the control benchmark catchment (with no human activity of interest) [11]. This approach is a classical method in hydrology by which the impact of disturbance on the catchment hydrology can be detected [13,14]. The climatic variability can be assessed by considering the same time period [15]. The "pre-post-disturbance" approach (iv) can make a before and after comparison between hydrological droughts of specific disturbances (pre-post-disturbance) [16,17] (e.g., Liu et al., 2016). In this approach, it is difficult to compare two different periods and make a distinction between the human effect and climate variability and climate change [17]. The "large-scale screening" approach (v) can quantify different drivers of hydrological processes through large samples [18] or comparative hydrology [8]. The negative points of this approach are a need for too many catchments with long time series of hydrological variables and the type and degree of human changes in the hydrological system for each catchment should be available [11].

The aim of the current research was extending the "observation-modelling" approach as a step-wise methodology to assess different drought types in the human-influenced era, e.g., climate-induced, human-induced and human-modified droughts. In addition, quantifying alleviated or enhanced human-modified droughts is the second purpose of this study. Designing a methodology to identify the human-modified drought type is the motivation for the research. Therefore, the observation-modelling approach was utilized and extended to quantify the human effect on hydrological drought. The capability of the expanded methodology was assessed in two case studies in Iran, e.g., Kiakola and Eskandari (with major human intervention concerns) to answer three main questions as follows:

(i)     What are the main and dominant types of drought?
(ii)    How have human interventions changed drought types?
(iii)   How has human modification enhanced or alleviated modified droughts?

The paper was formed as follows. In Section 2, the case studies are explained in detail. The expanded approach and methodology are then described in Section 3. Next, the results of the quantification of different hydrological droughts in the Anthropocene are shown in Section 4. Finally, the discussion and conclusion are presented in Sections 5 and 6, respectively.

## 2. Study Areas

Two Iranian catchments have been chosen, because these areas experience climate-induced droughts and usually have to cope with human interventions that cause the drought characteristics to change, either in a positive or in a negative way.

### 2.1. Kiakola Catchment

This catchment with an area of about 2100.9 km$^2$ and elevation between 9–3991 m above sea level, is located in 35°44′ to 36°19′ latitudes and 52°35′ to 53°23′ longitudes, in the Mazandaran province, in northern Iran (Figure 1, up). The type of precipitation (about 700 mm per year) is snow and rainfall in the mountainous and plain sectors, respectively. Most precipitation occurs in winter. July, with 27 mm of rainfall, is the driest month and the most precipitation, with an average of 177 mm falls in October. In the Mazandaran province, there are too many wetlands which have been exposed to drought because of climate change and anthropogenic activities. The main anthropogenic activities are land use change and over-extraction of surface and sub-surface water resources. In the past two decades, the forest area has decreased and Kiakola catchment as a part of the Talar watershed has been one of the most affected areas due to rapid urbanization and population growth. In addition, high demand for fuel and food, increasing the smuggling of timber and livestock grazing, development of road and cities, mine exploitation and factory construction have been recognized as the main human activates in this area. Changing the forest to grassland and grassland to residential sites has become one of the most major concerns in this area. The land-use maps of 2000 and 2011 have shown that the area of rangelands and forest lands has decreased and been converted into the residential, garden and agricultural land (for rice cultivation) [19,20]. The demand for rice production in Iran is increasing, while rice cultivation requires a high level of water depth at every growth stage. However, the occurrence of hydrological drought can alter the amount of rice which is needed to supply food demand [21].

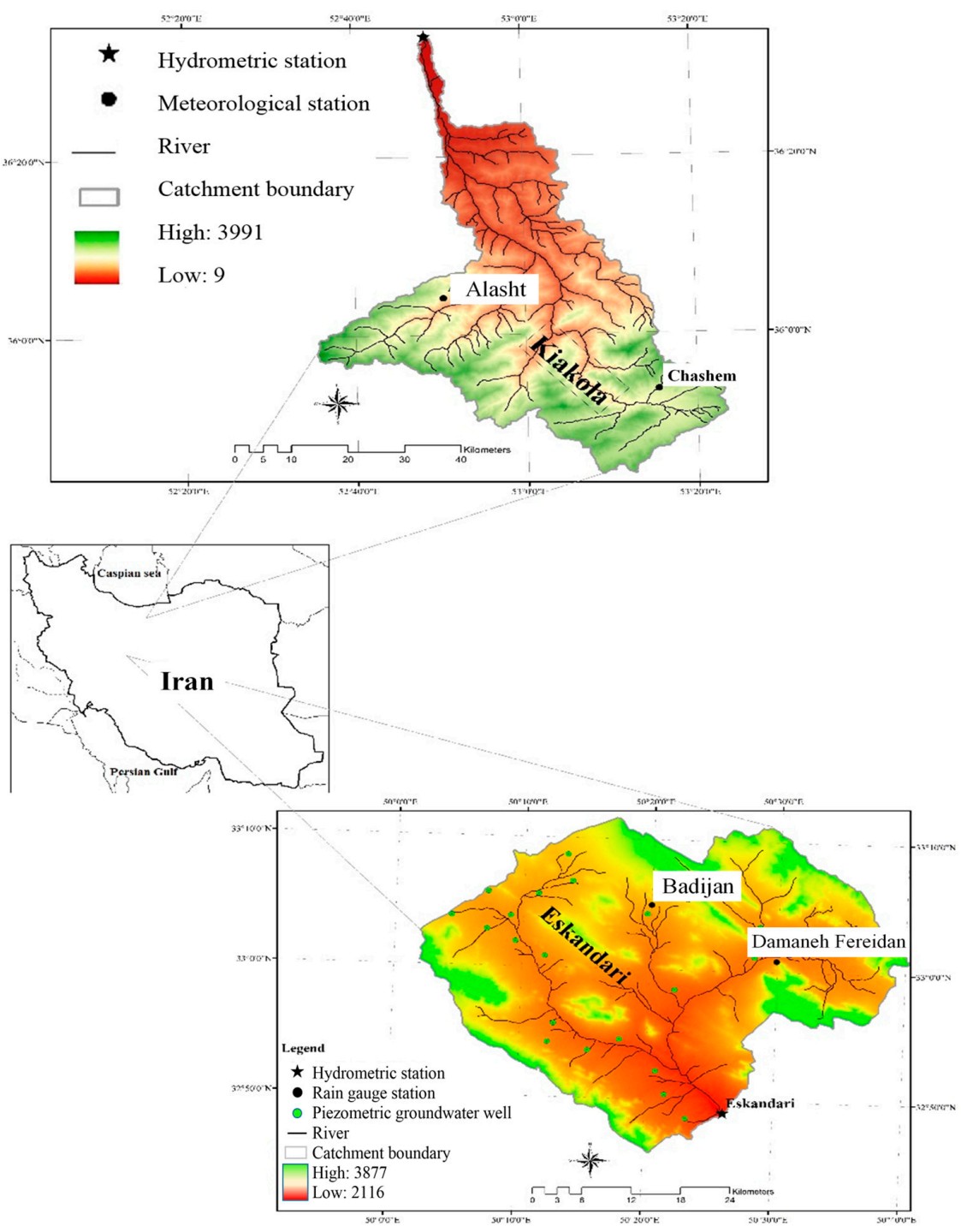

**Figure 1.** Location of study areas in Iran.

## 2.2. Eskandari Catchment

The Eskandari catchment with an area of 1649 km$^2$ and an elevation about 2116–3877 m above sea level is located in the upstream sector of the Zayandehrood dam basin, Esfahan province, Iran, at longitudes and the latitudes of 50°2′–50°41′ and 32°12′–32°46′, respectively [22] (Figure 1). The climate is classified as BSK (B (main climate): arid, S (precipitation): steppe, K (temperature): cold arid) based on the Köppen–Geiger climate classification [23]. The type of precipitation is different in the mountainous (snow) and plain (rainfall) sectors and the catchment receives the majority of its annual precipitation (about 420 mm per year) in autumn, winter and early spring (November–March).

The most important river in this catchment is the Pelasjan River and the flow discharge of the river is monitored at Eskandari hydrometric station.

During the past decades, water resources management in the Zayandehrood river basin has been a critical issue because of increasing human population, industry, high agricultural water use and repeated droughts. It has been reported that hydrological and meteorological extreme events are common events in this basin [24]. The Zayandehrood River, as the main element of human development in the central part of Iran, dries up seasonally, which leads to vast pressure on the urban populations, agriculture and industries [25]. The main anthropogenic pressure on the hydrological system in this catchment have been reported as increasing population growth [26], intensive pressure and over-exploitation of groundwater storage, increasing agricultural and residential areas (during 1997–2015) and decreasing pasture and forest lands [27]. During the past decades, frequent droughts have decreased the Zayandehrood Dam's lake to less than 150 MCM (million cubic meters) [28] and resulted in dryness of the Gavkhooni swamp. The Gavkhooni swamp is the habitat for a large number of flora and fauna, including many bird species, and the swamp is capable of refining water. Unfortunately, the degradation of the swamp has started due to a decrease in the quantity and quality of incoming flows [29–31]. The Eskandari catchment and the Pelasjan River, as one of the most important branches of the Zayandehrood River, upstream of the Zayandehrood's dam basin, have a significant role in the restoration and preservation of the Gavkhooni swamp. The inflow fluctuation has also been substantially intensified due to frequent droughts and water shortage, particularly downstream [32].

## 3. Materials and Methods

The extended methodology to quantify anthropogenic droughts is shown in Figure 2. Here, a concise description of the applied step-wise methodology has been presented. In the current research, step 5 is novel and an additional step to the presented approach by Van Loon and Van Lanen (2013) [33]. In this approach, hydrological modelling is needed for naturalizing hydrological variables and producing a long time series of hydrometeorological variables, e.g., discharge and groundwater storage time series [34]. The existing trend in hydrometeorological variables is detected by the Mann–Kendall test (Step 1). The change point in the time series of the variable of interest is also characterized by Pettit's test (Step 1). The change point can divide the whole study period into the "natural" and "disturbed" periods. For the natural period, the hydrological model will then be calibrated and validated (Step 2). The calibrated model will be applied to the disturbed period by which the flow discharge of the disturbed period will be naturalized. For drought analysis, the threshold will be calculated for the undisturbed period (natural) and will be applied to the disturbed period (Step 3). Next, by making a comparison between three groups of variables, including observed time series, naturalized time series and the threshold of the variable of interest, different types of natural and anthropogenic droughts will be quantified in the study period. Finally, in the last step, which is the innovative part of the current research, alleviated (positive) or enhanced (negative) modified droughts will be distinguished through anomaly analysis among the observed and naturalized time series for specific modified events. Further descriptions of the five steps are presented in the following sections.

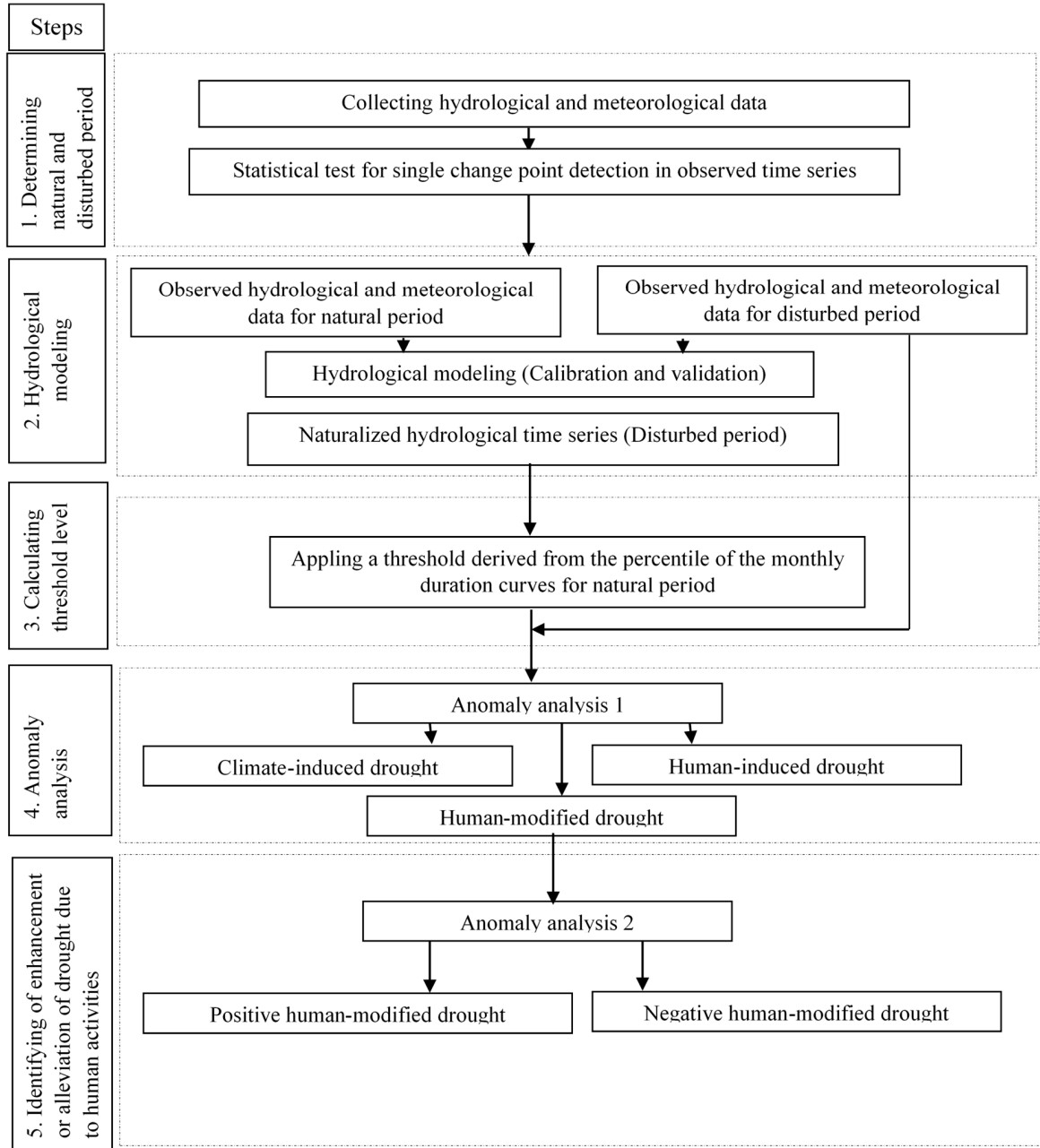

**Figure 2.** The expanded "observation-modelling" approach to quantify droughts in the Anthropocene.

## 3.1. Required Data

### 3.1.1. Observation Hydrometeorological Data

In this study, the observation time series of daily flow discharge, rainfall and temperature were needed for both Eskandari and Kiakola catchments. For the Kiakola catchment, the observed hydrometeorological data were available for the period of 1976–2013. The data have been provided by the Iran Water Resources Management Company, I.R. of the Iran Meteorological Organization and the Regional Water Company of Mazandaran and collected from Kiakola, Alasht and Chashem rain gauge stations and Kiakola hydrometric station. In this area, we did not perform groundwater drought analysis because the data on groundwater levels were not available.

For the Eskandari catchment, the discharge, rainfall and temperature data were available for the period of 1976–2014. In addition, there were 22 groundwater wells with an appropriate distribution

across the catchment and long time groundwater data (1983–2014) for each well. The daily rainfall, discharge and temperature were gathered for the Eskandari and Damaneh-Fereidan rain gauge stations, Badijan climatology station and Eskandari hydrometric station from the Iran Water Resources Management Company, I.R. of the Iran Meteorological Organization and the Regional Water Company of Isfahan.

### 3.1.2. Modelled Data

Hydrological modelling was performed to produce modelled data. For both catchments, the modelled data were naturalized discharge and groundwater storage time series for the disturbed period. The hydrological model was chosen based on the three main factors, e.g., flexibility, capability and a moderate amount of input data because of limitation of access to hydrometeorological data in Iran. The Hydrologiska Byråns Vattenbalansavdelning (HBV) model is a conceptual, semi-distributed rainfall-runoff model which has a simple and flexible structure. The ability of the model for rainfall-runoff modelling has been evaluated for different climatic conditions and the model is not a data-intensive model.

### 3.2. Step 1: Subdivision of the Total Period in Disturbed and Natural

The effects of human interventions and climate variability on hydrometeorological variables can be assessed according to differences in time series from both undisturbed (natural) and disturbed periods. At first, the non-parametric Mann–Kendall trend test was performed to detect possible trends in the observed time series, e.g., rainfall, discharge, temperature and potential evapotranspiration by which the human impacts on the variable of interest can be recognized [34–37]. In addition, the Pettit's test [38] was utilized to detect the change-points in long time series of observed hydrometeorological data (Step 1, Figure 2).

### 3.2.1. Mann–Kendall Trend Test

The Mann–Kendall test was utilized in order to detect trends in hydrological and meteorological time series [39,40]. In the Mann–Kendall test, $H_1$ implies that the data follow a monotonic trend and are not identical. $H_0$ as the null hypothesis, mentions that the data come from a population with an independent realization and are identically disturbed. In addition, an upward and downward trend can be determined by the positive and negative value of trend magnitude, respectively [35]. In hydrology and climatology, the Mann–Kendal test has been widely utilized to test for randomness against the trend and more information about the test is available in previous studies [41–45].

### 3.2.2. Pettitt's Test

The non-parametric Pettitt's test approach [38] was applied to detect the point of change in time series that separates the whole period into the undisturbed and disturbed periods. Pettit's test can identify whether two selected data samples belong to the same population based on the mean value of time series and the Mann–Whitney statistic. This test was applied to the time series of rainfall and flow discharge for both the Eskandari and Kiakola catchments. As a result, the study period was divided into the disturbed and natural periods with and without significant human interventions, respectively.

### 3.3. Hydrological Modelling for Naturalizing Hydrological Variables Time Series

In different parts of Iran with high levels of human interventions in the hydrological cycle, the river discharge data were influenced and needed to be naturalized. In addition, the long time groundwater storage data were not available, which were needed for groundwater drought analysis. Hence, the hydrological modelling was done by the HBV rainfall-runoff model to meet the basic research needs and requirements [33,46–48]. Through hydrological modelling, the HBV model was

calibrated and validated for the natural period and applied to the disturbed period by which the flow discharge and groundwater storage of disturbed period were naturalized.

### 3.3.1. HBV Rainfall-Runoff Modelling

The HBV (Hydrologiska Byråns Vattenbalansavdelning) model, has four different routines, e.g., snow, soil, response and routing. In the snow routine, the snow accumulation and melt are calculated by precipitation and temperature data as an input and degree-day method. The soil moisture routine can calculate the basin's degree wetness and merge interceptions and storage of soil moisture. The excess water from the soil moisture routine is transformed to discharge by the runoff response routine. Recharge is transformed into discharge by the two linear reservoirs in series in the STANDARD respond routine. While the DELAY response routine is composed of two parallel linear reservoirs. In addition, the channel routing will be calculated using a triangular weighting function [49–51]. Further description of the model can be found in related research [52–56]. It is recommended to use the DELAY response routine instead of the STANDARD routine in areas with slowly responding and deep-groundwater conditions [52,53].

### 3.3.2. Calibration and Verification of the HBV Rainfall-Runoff Model

In order to illustrate the methodology, the HBV model with the DELAY response routine was utilized with observed meteorological data in both catchments. The precipitation data was averaged using Thiessen polygons and the FAO Penman–Monteith was chosen to compute the potential evaporation [57]. The primary parameter estimation was calculated by model warm-up for both areas because of considerable inter-annual climate variability. The natural period that was specified by Pettit's test was then divided into the two periods for model calibration and verification. Finally, the calibrated model (imposed to natural period) was applied to the disturbed period and the flow discharge was naturalized. The main focus of modelling was allocated to precisely calculate low flow values through a genetic calibration algorithm [52]. The possible ranges of different parameters were defined by previous HBV applications in other studies [56,58]. Further description of the calibration of the HBV model by the genetic algorithm can be found in studies related to HBV modelling [55,58].

The model's performance was measured using the Nash–Sutcliffe efficiency of the logarithmic modelled and logarithmic observed discharge values (lnNSE). The peak flow values are able to alter the NSE and the $R^2$ (coefficient of determination) due to the lower sensitivity of these indices to low flow [59,60]. The $R^2$ is also only a function of the correlation between the modelled and observed data. The peak values of flow discharge can be flattened by transformation into the logarithmic values. As a result, the low flow and flattened peak values are approximately retained at the same level and the low flow value will be more effective on lnNSE sensitivity [61]. The acceptable rage of NSE has been declared equal to or more than 0.5 (NSE ≥ 0.5) [61,62]. Table A1 shows the value ranges of parameter and the structure of the HBV model [56,58].

### 3.4. Step 3: Calculating the Threshold Level

In the current study, the threshold level method was utilized for the analysis of hydrological drought [63,64]. According to the principles of this method, an anomaly can be considered as a drought event when the variable is below the threshold. The drought event will end when the variable exceeds the top of the threshold. Drought in the dry season due to water deficit in the wet season can be simulated precisely by application of the variable threshold, because the variable threshold is able to consider the seasonal pattern [65]. In the current research, the threshold level derived from the natural period (without significant human interventions) and then imposed to all time series. The variable monthly thresholds were also smoothed using a moving average of 30 days. For both catchments, the related droughts were then pooled through an inter-event approach and time period of ten days [66,67] and all droughts with duration less than fifteen days were omitted.

### 3.5. Step 4 and 5: Anomaly Analysis

In the Anthropocene, due to the intricate interplay between meteorological anomalies, land surface processes and human interventions in the state of the hydrological cycle, the catchment state is dependent on both natural and human processes. As mentioned in the introduction section, drought condition and propagation can be altered under natural and human processes. We live in an environment where natural processes and human interventions are interconnected in several ways by which drought propagation from meteorological to hydrological drought is affected by natural and human drivers (e.g., water abstraction, dam building, irrigation, land use change, etc.). Droughts in the Anthropocene have been categorized into natural drought or climate-induced drought and human droughts, e.g., human-induced drought and human-modified drought. Human-modified droughts occur when climate-induced droughts (natural drought) are exacerbated or alleviated by the aforementioned anthropogenic activates [68]. The different anthropogenic drought types are schematically demonstrated in Figure 3 (Please note that the CID, HID, and HMD are abbreviations of climate-induced drought, human-induced drought and human-modified drought, respectively). The solid brown line, dashed black and dashed purple horizontal lines present the observed time series, simulated time series and threshold, respectively.

When the naturalized time series (dashed black line) ($X_{[nat_{i,t}]}$) is under the defined threshold ($\tau_{i,t}$) (dashed purple horizontal line), a natural climate-induced drought happens because of climate variability [6,7]. The climate-induced drought is defined as [69]:

$$Climate - induced\ drought: \delta_{i,t} = \begin{cases} 1\ X_{[nat_{i,t}]} - \tau_{i,t} \leq 0 \\ 0\ X_{[nat_{i,t}]} - \tau_{i,t} > 0 \end{cases} \tag{1}$$

in which the drought situation on time $t$ is indicated by the binary variable ($\delta_{i,t}$). A human-induced drought which is a kind of drought caused only by human influence may occur by human drivers (with no natural drivers) of drought ($X_{[nat_{i,t}]} - \tau_{i,t} > 0$) when the observed variable of interest $X_{[O_{i,t}]}$ time series (solid brown line) is below the defined threshold (dashed purple vertical line):

$$Human - induced\ drought: \delta_{i,t} = \begin{cases} 1\ X_{[nat_{i,t}]} - \tau_{i,t} > 0\ and\ X_{[O_{i,t}]} - \tau_{i,t} \leq 0 \\ 0\ X_{[nat_{i,t}]} - \tau_{i,t} > 0\ and\ X_{[nat_{i,t}]} - \tau_{i,t} > 0 \end{cases} \tag{2}$$

As a combination of human influence and climate variability, a human-modified drought event occurred as a simultaneous anomaly in the naturalized ($X_{[nat_{i,t}]}$) and observed time series ($X_{[O_{i,t}]}$) at time $t$ as follows:

$$Human - modified\ drought: \delta_{i,t} = \{1\ X_{[nat_{i,t}]} - \tau_{i,t} \leq 0\ and\ X_{[O_{i,t}]} - \tau_{i,t} \leq 0 \tag{3}$$

A human drought event may be exacerbated or alleviated compared to the natural state. Therefore, positive-human modified drought happens when climate-induced drought is alleviated because of anthropogenic activates. A negative human-modified drought is related to the conditions when a natural drought is exacerbated by human interventions. By comparing $X_{[O_{i,t}]}$ and $X_{[nat_{i,t}]}$ and the threshold, human-modified drought types can be calculated as follows:

$$X_{[nat_{i,t}]} - \tau_{i,t} \leq 0\ and\ X_{[O_{i,t}]} - \tau_{i,t} \leq 0\ if \begin{cases} X_{[O_{i,t}]} < X_{[nat_{i,t}]}\ Negative \\ X_{[O_{i,t}]} \geq X_{[nat_{i,t}]}\ Positive \end{cases} \tag{4}$$

Equations (2)–(4) are formulated base on the existing knowledge [6,7]. In other research [6,7,70–73] more information was presented on human-induced, climate-induced and human-modified droughts and their drivers, impacts and modifiers in the Anthropocene. In the current research, drought analysis was done by Matlab programming. Various types of threshold can be utilized by applying the described methodology.

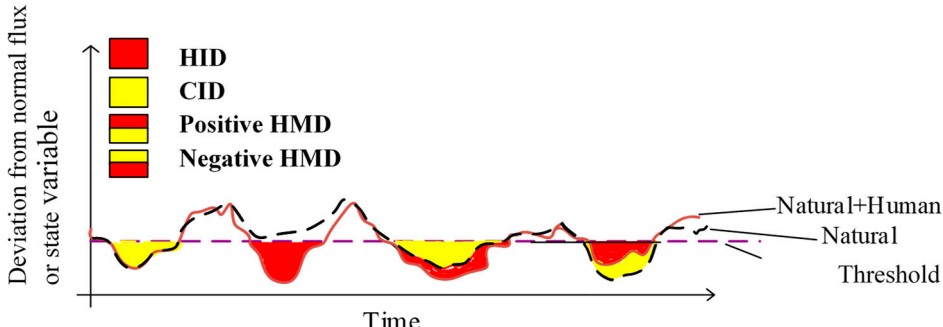

**Figure 3.** Schematic outline of a hydrological drought types in the Anthropocene (adjusted extended from Van Loon et al., 2016 [6]). A fixed threshold is used for simplicity rather than a variable threshold. CID, HID and HMD are abbreviations of climate-induced drought, human-induced drought and human-modified drought, respectively. The solid brown line, dashed black and dashed purple horizontal lines present the observed time series, simulated time series and threshold, respectively.

## 4. Results

### 4.1. Subdivision the Period

The Mann–Kendal test's results (Section 3.2.1) on the hydrometeorological time series are shown in (Table 1). The results of Mann–Kendal tests in Kiakola showed that trends in rainfall and temperature were upward and significant. Trends in discharge and evapotranspiration were significant downward and non-significant upward, respectively. For the Eskandari catchment, the trend in rainfall and evapotranspiration time series were non-significant and upward. While, trends in temperature and discharge were significant upward and downward, respectively. The results of the Mann–Kendall trend test in both catchments indicated the influence of both human activities and natural drivers. However, due to the increase in temperature, the higher potential evapotranspiration could have had an expanded effect on the discharge.

**Table 1.** The Mann–Kendall test's results.

| Study Area | Type of Variable | Tau | p-Value | Kendall Test Positive Significance | Sen's Slope |
|---|---|---|---|---|---|
| Eskandari | Q | −0.18 | <0.0001 | 1% | $-6.88 \times 10^{-6}$ |
| | R | 0.03 | 0.043 | NS | $3.44 \times 10^{-5}$ |
| | T | 0.02 | <0.0001 | 1% | $7.61 \times 10^{-5}$ |
| | Eva | 0.01 | 0.017 | NS | $2.01 \times 10^{-5}$ |
| Kiakola | Q | −0.08 | <0.0001 | 1% | $-1.2 \times 10^{-4}$ |
| | R | 0.07 | <0.0001 | 1% | $1.26 \times 10^{-4}$ |
| | T | 0.05 | <0.0001 | 1% | $1.46 \times 10^{-4}$ |
| | Eva | 0.02 | 0.014 | NS | $2.19 \times 10^{-5}$ |

The significance trends are highlighted and NS means no significance.

The results of Pettit's test in the Kiakola catchment show that the change point in rainfall and discharge was occurred in September 2004 and May 1998, respectively. According to the Mann–Kendall test (significant upward trend in rainfall, significant downward trend in discharge and non-significant upward trend in evapotranspiration) and Pettit's test results, the natural and disturbed period were determined. The period of 1976–1998 introduced as natural period and 1998–2013 defined as disturbed period (Table 2, Figures 4 and 5). For the Eskandari catchment, the change point in the time series of rainfall and discharge was approximately detected in October 2001 and May 1996, respectively. According to the Pettit's tests results, the entire period has been divided into the 1976–1996 and 1996–2014 as natural and disturbed periods, respectively. This pattern in discharge time series might

have been caused by land use and crop pattern changes, and abstraction from surface water resources and groundwater in both areas. Land use change has substantial impacts on the state of a hydrological system. The quantitative changes in surface water resources have been identified as one of the most important effects of land use changes.

**Table 2.** Change point detection using Pettit's test.

| Study Area | Type of Variable | p-Value | Alpha | Change Point |
|---|---|---|---|---|
| Eskandari | Q | <0.0001 | 0.01 | May 1996 |
| | R | <0.0001 | 0.01 | October 2001 |
| Kiakola | Q | <0.0001 | 0.01 | May 1998 |
| | R | <0.0001 | 0.01 | September 2004 |

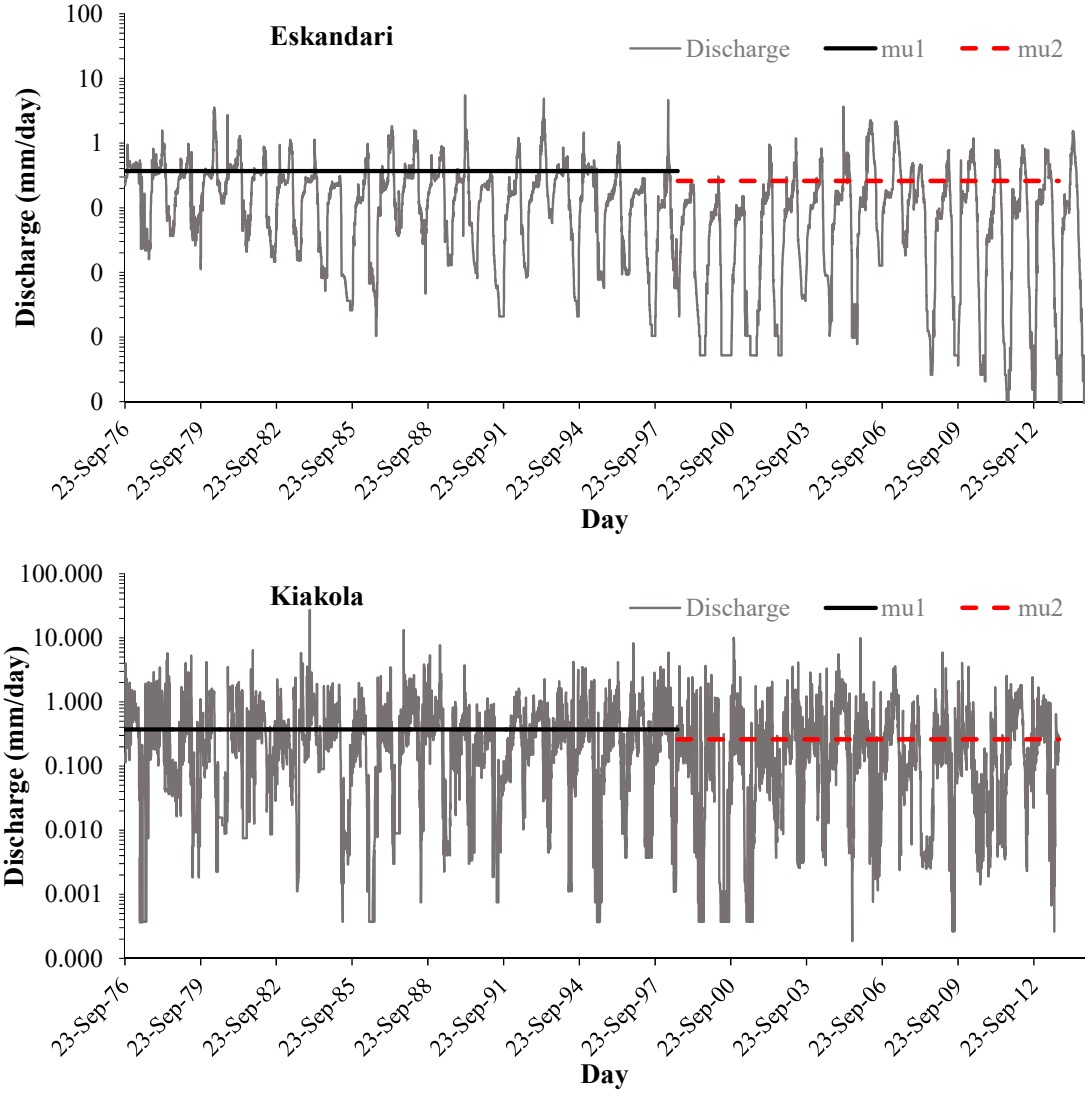

**Figure 4.** Pettit's test for discharge.

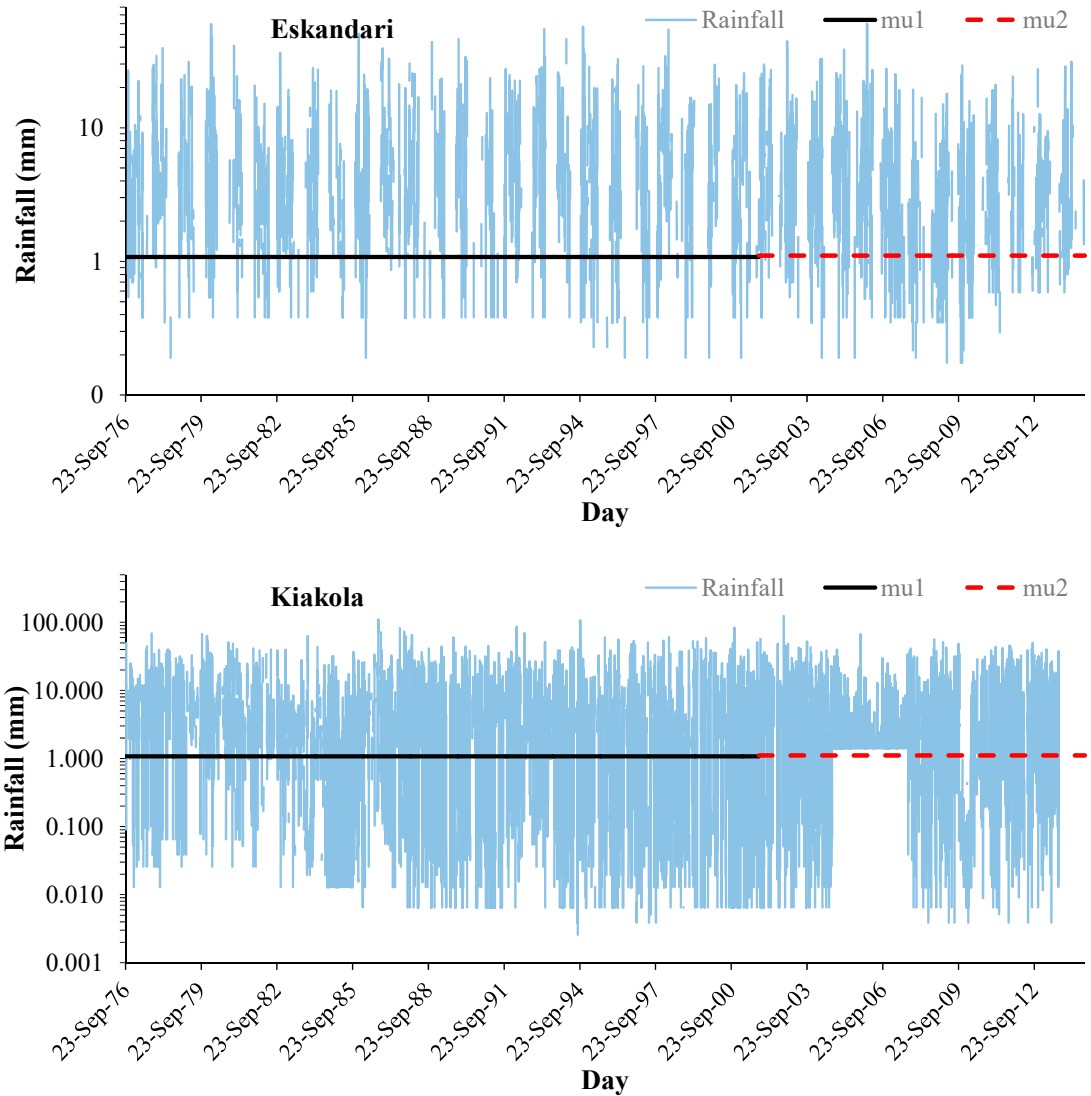

**Figure 5.** Pettit's test for rainfall time series.

To further examine changes in discharge in both catchments, the flow duration curves (FDCs) using monthly discharge records were plotted for the naturalized and disturbed period in both catchments. The FDCs for the disturbed period were generally lower than for the natural period implying that river flow has decreased due to human interventions (Figure 6).

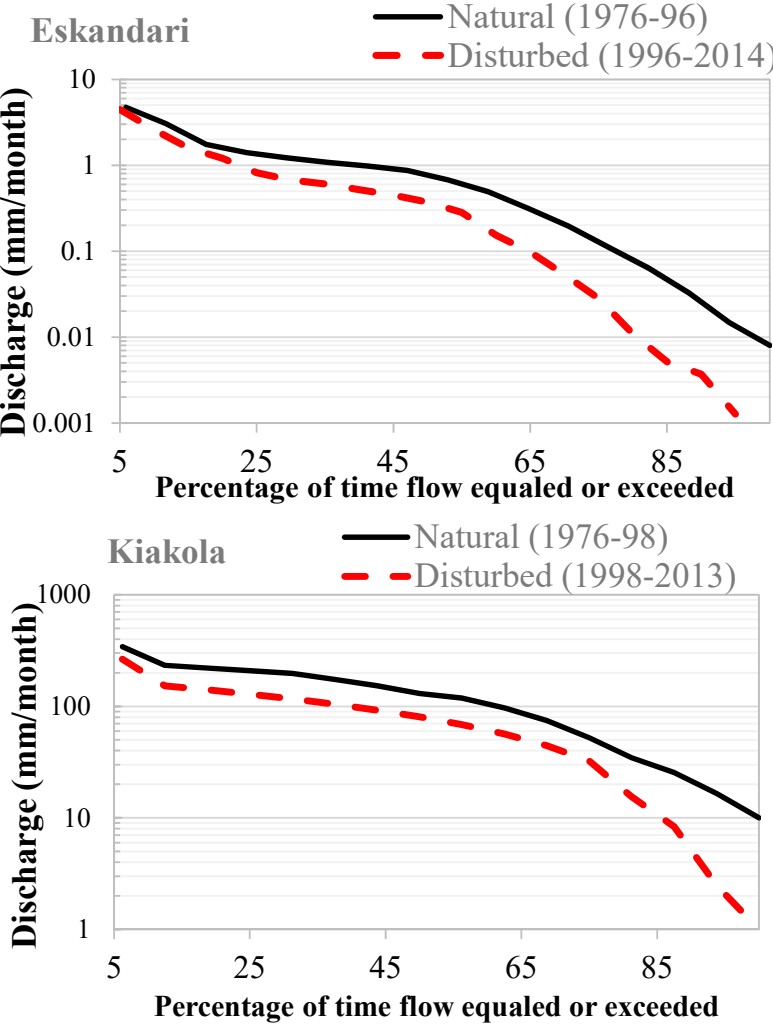

**Figure 6.** Flow duration curves of monthly discharge for natural and disturbed conditions.

*4.2. Obtaining the Naturalized Time Series through Hydrological Modelling*

The results of HBV modelling for both catchments are shown in Table 3 (for the natural period). The Nash–Sutcliffe efficiency values of the logarithmic data were about 0.5, which pointed to the acceptable performance for the natural period. The value of $R^2$ was more than 0.6 for model calibration and verification. Due to severe seasonal variation in Eskandari, the *lnNSE* of this area was lower than Kiakola.

The flow duration curves of the observed and simulated flow discharge of natural condition for both catchments are shown in Figure 7. However, in the calibration period, the model underestimated some peak discharge values and with lower *lnNSE* , the simulated flow discharge agreed reasonably well with the observed discharge time series. The HBV has acceptably simulated the inter-annual and seasonal variability in flow discharge. In addition, compared to the simulated time series, the observed flow discharge was slightly more peaky. The comparison among different values of percentile, e.g., 50th and 80th of the monthly and annual duration curves of the simulated and observed flow discharge are shown in Table A2. The results of this table in both catchments indicate that the inter-annual variation has been reproduced rather well and that observed and simulated flow discharge percentiles have shown reasonable agreement. In order to identify various drought types in the disturbed period, the simulated discharge was utilized as an estimation of the naturalized flow discharge. The HBV model simulated groundwater storage with $R^2$ equal to 0.4, which was not good but acceptable [34], because the groundwater level was extracted by imposing the coefficient of storage,

which was a constant value of simulated storage. In total, the ability of the HBV hydrological model for both catchments was acceptable in the natural period and pointed to a reasonable ability in the disturbed period to simulate the naturalized hydrological time series.

**Table 3.** Hydrological modelling results.

| Study Area | Periods | *NSE* | *lnNSE* | $R^2$ |
|---|---|---|---|---|
| Kiakola | Calibration (1976–1992) | 0.54 | 0.57 | 0.62 |
| | Verification (1992–1998) | 0.56 | 0.5 | 0.6 |
| Eskandari | Calibration (1976–1990) | 0.53 | 0.57 | 0.62 |
| | Verification (1990–1996) | 0.56 | 0.50 | 0.62 |

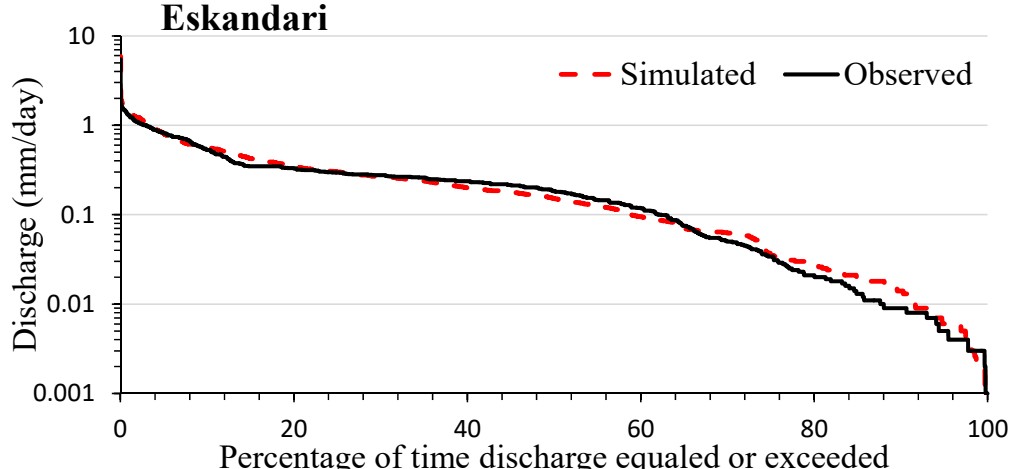

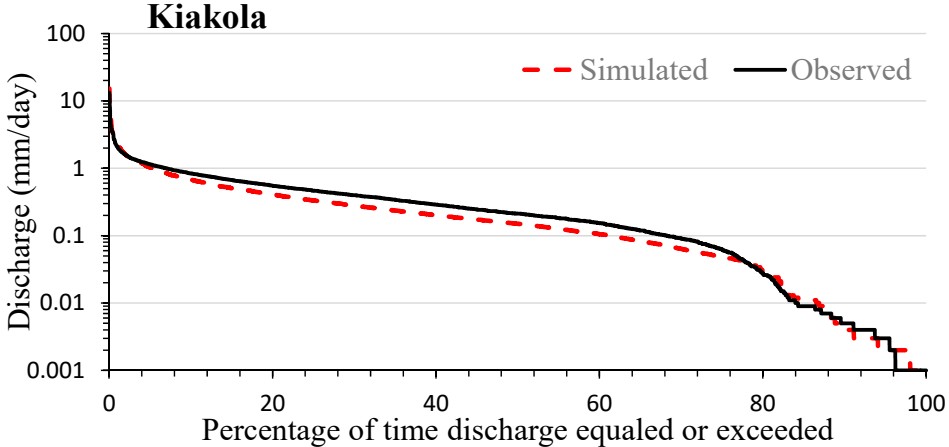

**Figure 7.** Flow duration curves for daily discharge.

*4.3. Step 3 and 4: Quantifying Drought Types in the Anthropocene*

After model calibration for natural period, the model was applied for the disturbed period. The discharge and groundwater time series were naturalized by HBV. In order to make a distinction between various drought types, a variable monthly threshold of discharge, which considers the seasonal variability [74], has been calculated from the 80th percentile of observed data for the natural period. The derived threshold was then applied to all periods in both study areas [71]. The human-affected and natural drought types were distinguished by making a comparison between the observed, modelled and the threshold of variables [33,48]. The results are shown in Tables 4–6.

Based on the definition of climate-induced drought and Equation (1) [69], climate-induced drought events were identified by linking the naturalized flow discharge and the threshold. For both Kiakola and Eskandari catchments during the study period, 34 and 41 climate-induced drought events were detected. Due to higher climate variability in Kiakola, the average, maximum water deficit and average duration of this type of event in the Kiakola catchment were lower than the dryer Eskandari catchment. For both Iranian catchments, the summary statistics of climate-induced drought are provided in Tables 4 and 5. In addition, the examples of natural drought are shown through hydrographs (Figure 8).

In addition, several severe human-induced droughts were quantified by connecting the threshold and the observed flow discharge data (Equation (2), Figure 9). Similar to the climate-induced drought, in the Eskandari catchments the human-induced drought events, were more frequent, severe (the average of water deficiency was about three times larger) and lasted longer (the average duration was about twice longer) than in the Kiakola catchment (Tables 4 and 5).

As a combination of human influences and climate variability, human-modified droughts were identified through comparing naturalized, observed discharge data and the threshold (Equation (3), Figure 10). Although both catchments were similar in the number of modified events, the modified droughts in Eskandari were 3.7 times more severe and 2.6 times longer than in Kiakola, which indicated that the dryer Eskandari catchment has experienced more severe and longer events.

In the Eskandari catchment, the number of climate-induced, human-induced and human-modified drought were 41, 45 and 39 events, respectively. Therefore, human-induced drought has been more frequent than other drought types (Table 4). Although modified droughts have been the least occurring event, the deficit volume, average and maximum duration of modified droughts were substantially higher than the other drought types.

The drought analysis results (Table 5) for Kiakola indicated that climate-induced droughts were more in number, longer in duration and more severe in average deficit volume than human-induced droughts. Although modified droughts were more frequent than other drought types, the average deficit volume was less than human-induced events. In comparison to climate-induced droughts, modified droughts were not the most severe events for the Kiakola catchment.

**Table 4.** Discharge drought characteristics for Eskandari (threshold = 80th percentile).

| Type of Drought | No. Drought | Duration (day) | | Deficit Volume (mm) | | Human Effect (%) | |
|:---:|:---:|:---:|:---:|:---:|:---:|:---:|:---:|
| | | Maximum | Average | Maximum | Average | Negative | Positive |
| CID | 41 | 204 | 62 | 10.3 | 2.4 | - | - |
| HID | 45 | 192 | 72.9 | 9.2 | 2.4 | - | - |
| HMD | 39 | 306 | 115.2 | 21.9 | 4.1 | 96.9 | 3.1 |

**Table 5.** Discharge drought characteristics for Kiakola (threshold = 80th percentile).

| Type of Drought | No. Drought | Duration (day) | | Deficit Volume (mm) | | Human Effect (%) | |
|:---:|:---:|:---:|:---:|:---:|:---:|:---:|:---:|
| | | Maximum | Average | Maximum | Average | Negative | Positive |
| CID | 34 | 173 | 42.6 | 20.3 | 2 | - | - |
| HID | 31 | 115 | 32.7 | 3.5 | 0.7 | - | - |
| HMD | 40 | 177 | 43.9 | 9.8 | 1.1 | 88.9 | 11.1 |

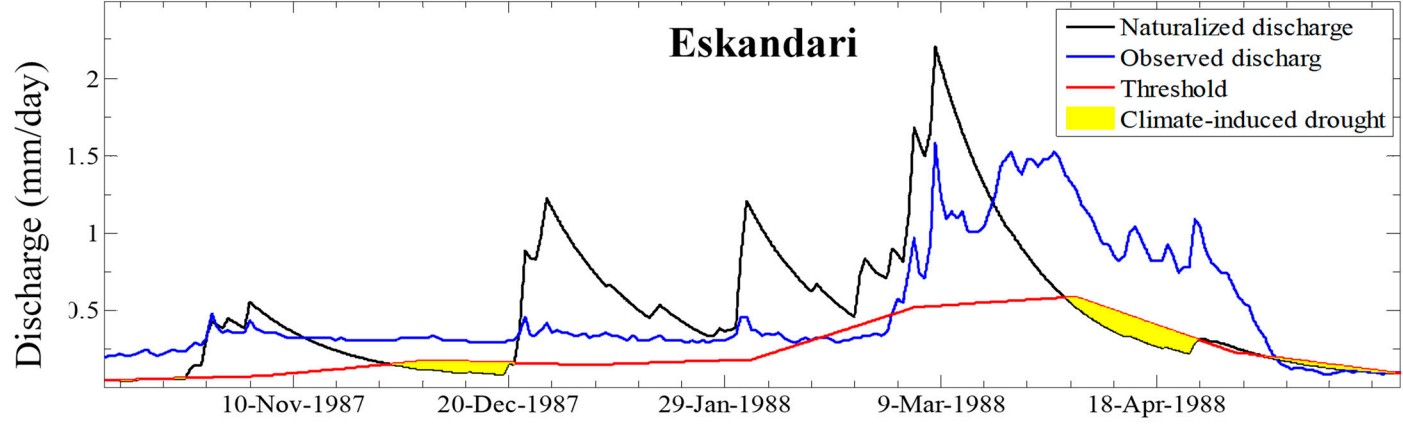

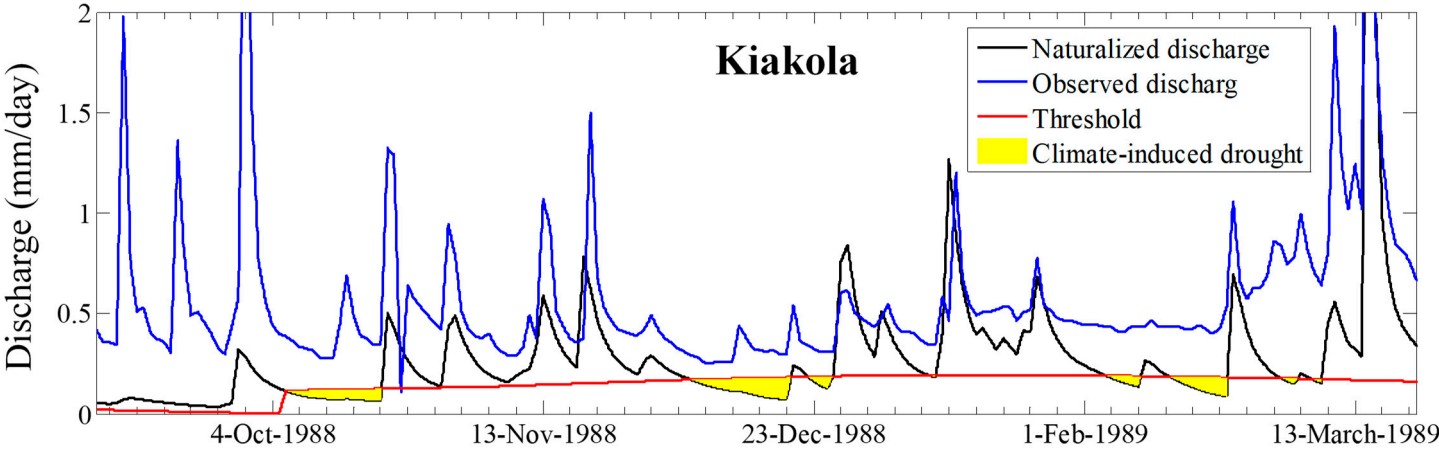

**Figure 8.** Examples of climate-induced droughts.

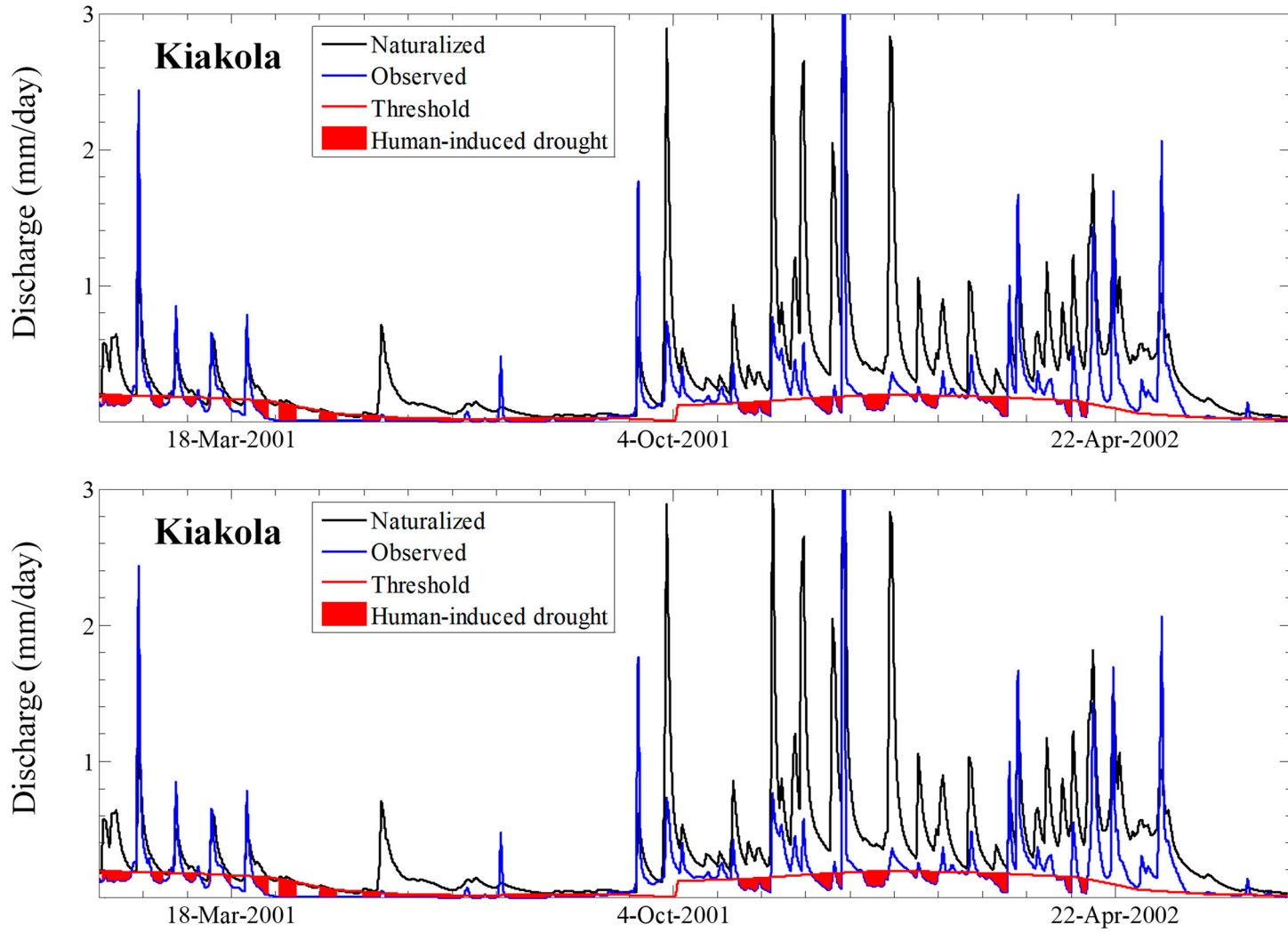

**Figure 9.** Examples of human-induced droughts.

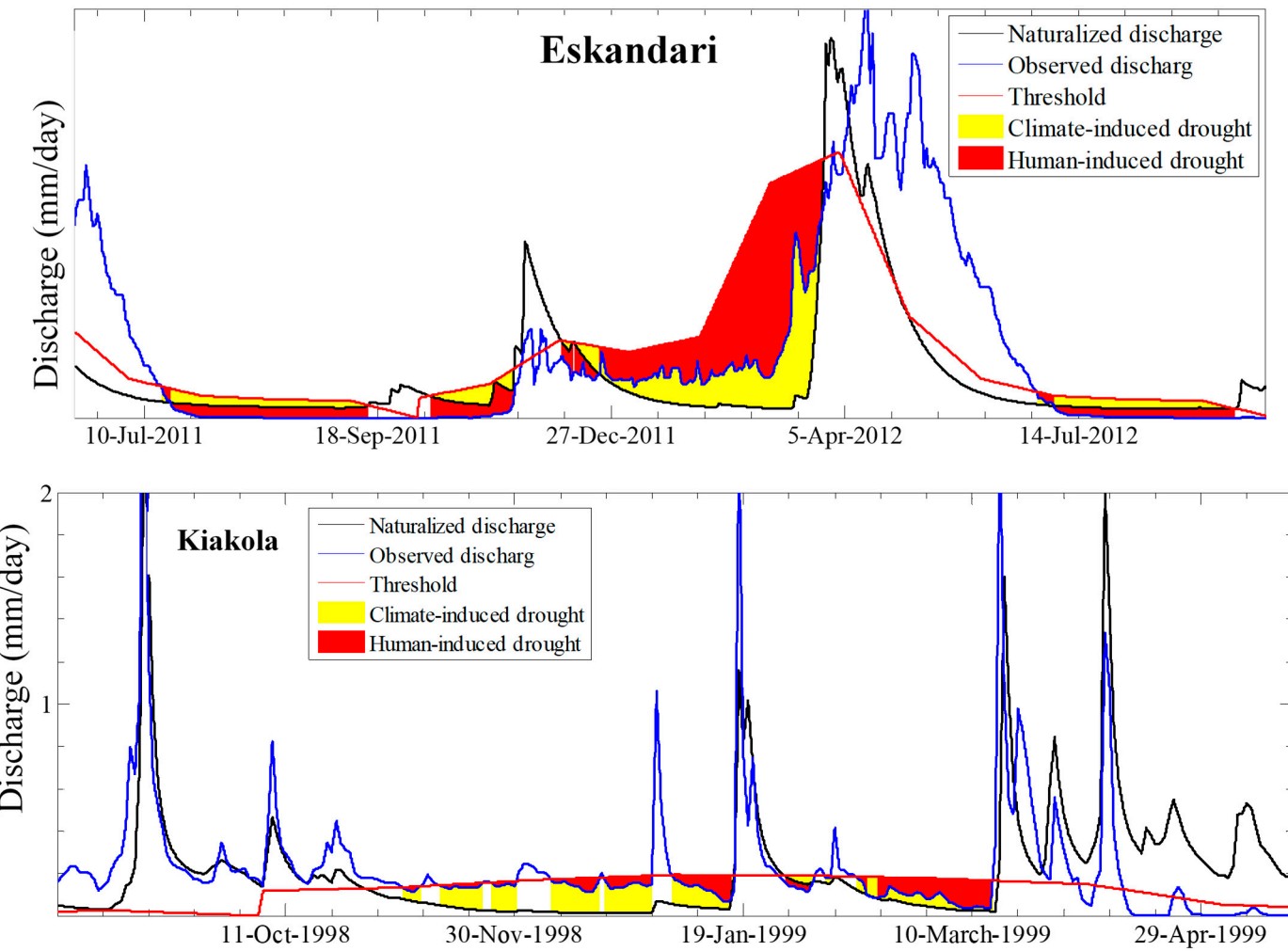

**Figure 10.** Examples of human-modified droughts.

Groundwater drought characteristics in the Eskandari catchment was shown in Table 6. The results indicated the number of groundwater droughts was less than the flow discharge droughts in the river but the duration of the modified groundwater events, with a maximum duration of 196 months, which has continued for more than 16.3 years, was notable (Figure 11). In the Eskandari catchment, the groundwater resources have not been able to recover for more than 16.3 years, due to long-lasting overexploitation and the severe human impact on groundwater storage.

**Table 6.** Groundwater drought in Eskandari (threshold = 80th percentile).

| Type of Drought | No. Drought | Duration (Month) | | Maximum Deviation (m) | | Human Effect (%) | |
|---|---|---|---|---|---|---|---|
| | | Maximum | Average | Maximum | Average | Negative | Positive |
| CID | 6 | 23 | 9 | 1.5 | 0.64 | - | - |
| HID | 6 | 100 | 23.4 | 14.6 | 7.2 | - | - |
| HMD | 6 | 196 | 37.3 | 15.6 | 3.8 | 100 | 0 |

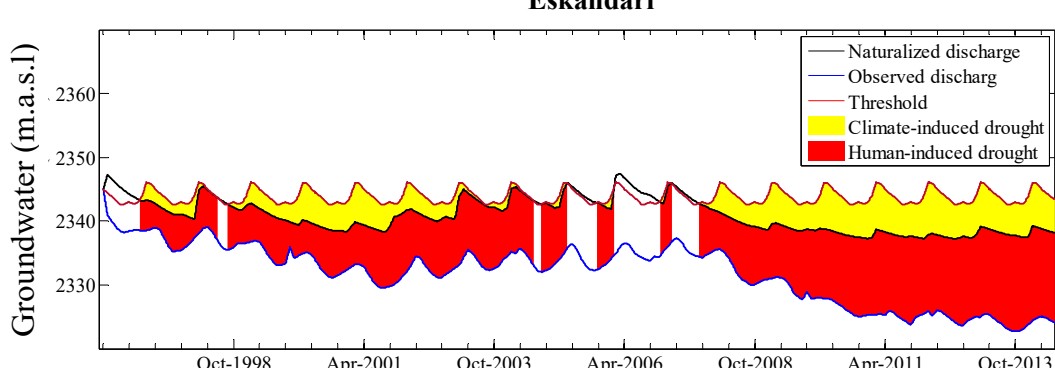

**Figure 11.** Groundwater drought types in the Anthropocene for Eskandari.

### 4.4. Step 5: Quantifying Negative and Positive Human-Modified Drought in the Anthropocene

An anomaly analysis (anomaly analysis 2, Figure 2) has been done based on Equation (4) to make a distinction between negative and positive modified droughts. In this way, the enhancing or alleviating effects of the human drivers of drought have been evaluated in the new final step in the presented approach.

The results of Tables 4–6 for both catchments revealed that more modified drought events were categorized as negative ones. In the Kiakola and Eskandari catchments, about 89% and 97% of droughts were classified as negative modified droughts, respectively. In addition, as a combination of the human pressures and natural drivers of droughts on groundwater storage (Figure 11), all groundwater droughts have been identified as negative modified droughts. Although both study areas were different in climatic condition, both were almost the same in experiencing longer and more severe negative modified droughts than positive ones because of the negative pressure of human activities on the hydrological system.

## 5. Discussion

The current study presented a methodology by which the enhancing or alleviating effects of human drought drivers (Figure 2) have been evaluated through extending the "observation-modelling" approach presented by Van Loon and Van Lanen (2013) [6,33,48]. To demonstrate the extended approach and quantify positive and negative modified events, two Iranian catchments were chosen as examples with notable human intervention and different climatic conditions. For drought analysis, the monthly-variable thresholds were identified (Step 3) and anomaly analysis performed (Step 4). In this section, the application of other thresholds in drought specification and the portion of positive and negative modified droughts (Step 5) have been assessed by applying various percentiles, e.g., the 50th,

70th and 90th. The 90th and 50th values of thresholds were the lower and higher percentiles, respectively. The statistical summary of the results is shown in Table 7. Applying different thresholds by various percentile values can alter the values of modified droughts. The results showed that in comparison to the 50th and 70th percentiles, the application of the 90th percentile threshold has created fewer modified droughts events with lower deficit volumes and shorter durations. The percentage of positive and negative modified droughts were not altered significantly by imposing various threshold and values of negative modified events were computed about 78–99%. The portion of positive modified groundwater events was altered between 0–17.9%, which was also very insignificant.

The obtained results showed that during the study period, human activities have a negative influence on the hydrological system in the Iranian catchments. In Eskandari, the positive and negative modified droughts happened in the wet season (November–March) and dry season (summer), respectively (Figures 12 and 13). The positive events were also not considerable events in terms of the number of events, deficit volume and durations. Hence, the system was not able to recover from severe negative events during the wet season. To supply the water demand during the dry season, the abstraction of groundwater and surface water resources was increased. Therefore, the most severe natural droughts, i.e., climate-induced events, were turned into negative modified events in the dry season [75,76].

**Table 7.** Human-modified drought characteristics.

| Study Area | Type of Variable | Threshold (Percentile) | No. Drought | Duration (day) | | Deficit (mm) for Discharge/Maximum Deviation for Groundwater (m) | | Human Effect % | |
|---|---|---|---|---|---|---|---|---|---|
| | | | | Max | Average | Max | Average | Negative | Positive |
| Eskandari | Q | 50th | 57 | 1250 | 128.4 | 164.2 | 10.8 | 79.2 | 20.8 |
| | | 70th | 42 | 1041 | 126.3 | 76.8 | 5.7 | 90 | 10 |
| | | 90th | 33 | 267 | 116.7 | 13.2 | 3 | 99 | 1 |
| Eskandari | GW | 50th | 4 | 234 | 71.3 | 17.4 | 6.6 | 82.1 | 17.9 |
| | | 70th | 6 | 218 | 42.3 | 16.2 | 3.8 | 97.6 | 2.4 |
| | | 90th | 4 | 195 | 52.3 | 15 | 5 | 100 | 0 |
| Kiakola | Q | 50th | 75 | 189 | 46.6 | 24.1 | 3.6 | 78.8 | 21.2 |
| | | 70th | 48 | 179 | 43.8 | 13.9 | 1.8 | 89.1 | 10.9 |
| | | 90th | 32 | 146 | 35.1 | 4.7 | 0.4 | 95.9 | 4.1 |

As mentioned in the introduction section, different approaches have been developed to quantify human influences on hydrological drought, e.g., observed-modelled, upstream-downstream, paired catchment, pre-post-disturbance, and large-scale screening.

At first, the "observed-modelled" framework was applied to quantify drought and water scarcity in the discharge and groundwater time series for the period of 1980–2000 in the Upper-Guadiana catchment in Spain [33]. Discharge, rainfall and temperature were input data for rainfall-runoff modelling by the HBV model. In another study, the performance of the approach was evaluated in three catchments in the Czech Republic (Svitata and BILINA catchments) and the Netherlands (Poelsbeek catchment) to quantify drought and water scarcity [48]. The lumped conceptual BILAN model and distributed physically-based SIMGRO were utilized as a hydrological model in the Czech Republic and the Netherlands, respectively. In 2016, this approach was applied to introduce and quantify three new concepts of watershed drought (WD), watershed water scarcity (WWS), and streamflow water scarcity (SWS) in Luanhe river basin, China [2]. The physically-based semi-distributed SWAT model was applied to model runoff time series. Daily rainfall and discharge data of 21 stations, temperature and land use maps (1970 and 1980) were the input data. This method is based on the availability of hydrological and meteorological data for the natural and disturbed periods, irrespective of the region. The observed-modelled approach had acceptable performance in all the aforementioned studies. Naturalization of the hydrological system can be performed by hydrological modelling as a central point of the current methodology. It is possible to choose different kinds of hydrological models in the framework, e.g., stochastic, lumped and conceptual or physically based models. Therefore, the kinds of necessary hydrometeorological data depend on the choice of hydrological model. At the global

scale, satellite hydrological data can be utilized. However, the high uncertainties of these kinds of data should be considered. Naturalization of the disturbed period time series is challenging and is extremely dependent on the precise calibration of the model, the regionalization methods and data about type and degree of human interventions with a suitable resolution.

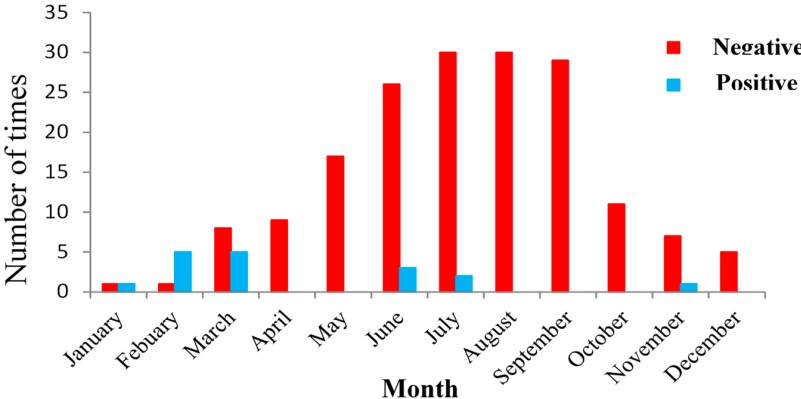

**Figure 12.** The distribution of modified droughts during a year for Eskandari (the 80th threshold value).

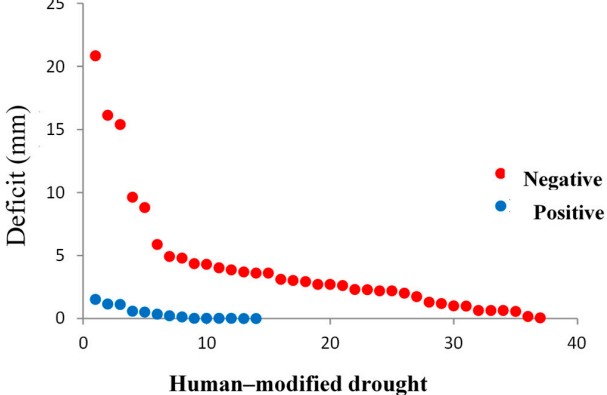

**Figure 13.** Deficit volumes of modified events for Eskandari (the 80th threshold value).

Uncertainty is an intrinsic part of any hydrological modelling. In addition, the quality and quantity of hydrometeorological data of arid and semi-arid regions have been the biggest concern of modelling. In the selected case studies, long time series of observed hydrometeorological data were not available. The spatial distribution of rain gauges has also likely impacted on the modelling. Besides that, security issues and water conflicts and chaos that have occurred over the past years have led to a reduction in the availability of information in many parts of Iran. As precise quantitative or qualitative information on water resource development projects in the Eskandari catchment was not possible to provide because of security issues. In the current research, we tried to minimize the uncertainty to an acceptable level by using a genetic calibration algorithm (GAP optimization) to find an optimal parameter set. Selection of the threshold level type is a challenging issue. In arid and semi-arid regions, most of the time the rainfall is zero. In addition, the discharge time series contains considerable zero or close to zero values. Therefore, it is possible to detect extended dry periods for several months [77–79]. Hence, it is possible to utilize the threshold of 50th percentile to be safe of a threshold of zero [80–82].

The results of the current research with the presented methodology for two Iranian catchments has corresponded well to drought analysis during (2001–2009) in Australia. The information of climate (ENSO, STR, PDO, and IOD indexes), water resources (AWRA hydrological model), economic factors, and remote sensing data were used to quantify "Millennium Drought" [83]. The previous study has also demonstrated that the deficit of negative modified drought has increased more the 50% because of the over-abstraction from underground water resources [10].

Whether the occurrence of modified drought was dependent on the purpose and reservoir management has been discussed [10]. The effect of reservoirs on hydrological drought in the Huasco basin in northern Chile has been assessed through the upstream-downstream approach for a long-term period of (1965–2013) [12]. The approach was utilized for the observation and modelled streamflow time series to analyze the post-dam and pre-dam periods as disturbed and undisturbed periods, respectively. For a similar analysis, the naturalized and human-influenced scenarios were generated using WEAP (Water Evaluation and Planning) modelled data. For the same period, the natural differences were quantified between upstream and downstream using calculation of downstream changes relative to upstream. In addition, human changes during the disturbed period were calculated by accounting natural differences. The results showed that the data and methods for drought analysis in the Anthropocene need to be chosen carefully. As calculating the threshold for the undisturbed period as a reference period, made it possible to exclude the human impact from normal conditions. In addition, they showed how the two methods of the threshold level and standardized indices differed due to the differences between the methods and including or excluding the human changes in the "undisturbed" situation [12]. The "paired-catchment" approach was applied in the UK (Chemler catchment as benchmark and Blackwater catchment) and Australia (Cockburn catchment as benchmark and Cox catchment) to quantify the human influence on hydrological drought [11]. The results demonstrated that this approach has been able to quantify drought by observation data. However, they noted that the main limitation of the method was the uncertainty of observation and modelled data.

During the past decades, the Eskandari catchment has suffered from severe droughts especially during 1998–2001 [83]. Over half of the population in the Zayandehrood basin has been influenced by the extended drought in 1998–2001 [84]. During 1995–2011, land use changes as the main human activity have increased the areas of agricultural lands by about 11%. Therefore, the vulnerability of the hydrological system to drought has increased because of a decrease in the water flow. In 1997–2016, the areas of different land uses were also changed [28]. The residential and agricultural areas have increased from 3679.8 $km^2$ and 7602.8 $km^2$ to 8574.05 $km^2$ and 8636.8 $km^2$, respectively. Rangelands and forest areas have decreased from 26,041.8 $km^2$ and 1499.8 $km^2$ to 21,285.5 $km^2$ and 930.54 $km^2$, respectively [25]. Increasing irrigated areas during 1996–2000, crop pattern changes, and applying 20% conversion from wheat to rice cropping have increased water consumption and the risk of hydrological drought. In Eskandari, more than 60% of irrigation water was supplied through groundwater abstraction [85]. In this area, water use efficiencies of all irrigation systems were very low (34%). Between 1956 and 2006, the rate of population growth has increased by about 5.9%. Hence, domestic and industrial demands have increased and hydrological droughts have aggregated. The construction of new water resources developments (e.g., Chadegan Reservoir and Kohrang Tunnels, from 1953–2020), by which the average annual yield increased from 1200 MCM in 1996 to 1790 MCM in 2010, could not overcome the vulnerability to drought under management practices.

In the Kiakola catchment, between 1991 and 2013 forest lands were reduced about 14.9% (from 83,902.75 ha to 71,424.71 ha). The areas of irrigated land, dry agricultural lands, rangelands and residential areas have increased by 46.8% (from 1017.82 ha to 1493.82 ha), 31.1% (from 23,272.06 ha to 30,520.31 ha), 4.7% (95,826.45 ha to 100,307.5 ha) and 17.5% (from 1563.77 ha to 1837.72 ha), respectively [20]. In the Kiakola, a large portion of the water demand was supplied from groundwater (63.1%) and the rest was provided from surface water resources (36.9%). About 87.8%, 11.4% and 0.9% of groundwater resources were allocated for agriculture, drinking and industry purposes, respectively [86].

## 6. Conclusions

Adaption and mitigation of future severe droughts need clear insight into the natural and anthropogenic drivers of droughts and an understanding of how variability in human drivers impacts anthropogenic drought in positive or negative ways. Therefore, the quantification of human drivers of drought is the main requirement for answering these questions. Subsequently, to address these research

gaps, the aim of the current research was to quantify the impacts of natural drivers and anthropogenic activity on hydrological drought. In the current study, the extension of the "observed-modelled" methodology by Van Loon and Van Lanen (2013) [33] was applied to separate and quantify different types of drought in the Anthropocene. Despite the focus of the current research on Iran, the methodology is not unique to Iran and can be applied in other areas around the world. The main concerns about human intervention in both Iranian catchments were land use change, crop pattern change, large groundwater abstraction and population growth. The expanded methodology was tested in a dry environment, because climate-induced droughts are frequent there and human interventions are common, likely causing modified drought. The results demonstrated that both study areas have experienced more severe negative and longer human droughts than natural droughts. The current study proved that the data of the two Iranian catchments are sufficient to illustrate the potential of the extended methodology. In a follow-up study, a larger set of catchments can be analysed to explore the differences between different climate settings and human influences. The application of the extended approach has implied that drought in our human-influenced era is interlocked with human activities and not simply natural drought due only to climate variability. Exploring and investigating the effect of human impacts on drought is very complicated because of divergent processes and confusion over the source of interrelated and complex processes [2,87]. Therefore, this approach enables water resources managers, policy makers and water scientists to find an effective way to combat drought types in the Anthropocene.

**Author Contributions:** E.K. and H.A.J.V.L. came up with the idea for the research and performed the analysis. E.K. and H.A.J.V.L. contributed equally to this research. H.A.J.V.L. provided continuous input and insight. E.K. wrote the manuscript with recommendations from all co-authors (H.A.J.V.L., H.R.M., A.M.N.). E.K. and H.A.J.V.L. helped to make the research structure reasonable.

**Funding:** This research was partly funded (H.A.J.V.L. and E.K.) by the European Commission, ANYWHERE project (Grant Agreement No. 700099), EU's Horizon 2020 research and innovation program (www.anywhere-h2020.eu).

**Acknowledgments:** The authors appreciate the research groups at Wageningen University and Tarbiat Modares University for their cooperation and discussions. This research is part of the Wageningen Institute for Environment and Climate Research (WIMEK-SENSE) and it supports the work of the UNESCO EURO-FRIEND-Water and the IAHS Panta Rhei program. The authors would also like to thank the editor and reviewers for their valuable comments.

**Conflicts of Interest:** The authors declare that they have no conflict of interest.

## Appendix A

**Table A1.** HBV model parameters and the value ranges (DELAY response routine).

| Parameter | Explanation | Unit | Initial |
|---|---|---|---|
| | Snow routine | | |
| TT | Threshold temperature | °C | (−2.5 2.5) |
| CFMAX | Degree-day factor | mm °C$^{-1}$ day$^{-1}$ | (1 10) |
| SFCF | Snowfall correction factor | - | (0.2 0.6) |
| CFR | Refreezing coefficient | - | (0 0.05) |
| CWH | Water holding capacity | - | (0 0.5) |
| | Soil moisture routine | | |
| FC | Maximum SM (storage in the soil) | mm | (70 300) |
| LP | Threshold for reduction of evaporation (SM/FC) | - | (0.3 1) |
| BETA | Shape coefficient | - | (1 2) |
| | Response routine | | |
| Alpha | Nonlinearity parameter | - | (0 1) |
| K1 | Recession coefficient | day$^{-1}$ | (0.02 0.2) |
| K2 | Recession coefficient | day$^{-1}$ | (0.0005 0.1) |
| | Routing routine | | |
| MAXBAS | Routing, length of weighting function | day | (1 10) |
| | Other | | |
| PART | the portion of the recharge which is added to the groundwater box | - | (0.5 1) |
| DELAY | period of delay | day | (1 50) |

**Table A2.** Comparison of 50th and 80th percentile of the duration curves of flow discharge and groundwater (Eskandari catchment).

|  |  |  | January | February | March | April | May | June | July | August | September | October | November | December | Annual |
|---|---|---|---|---|---|---|---|---|---|---|---|---|---|---|---|---|
| Eskandari | $Q_{sim}$ | 50% | 0.25 | 0.24 | 0.41 | 0.62 | 0.13 | 0.05 | 0.03 | 0.02 | 0.04 | 0.14 | 0.22 | 0.27 | 0.18 |
|  |  | 80% | 0.17 | 0.2 | 0.33 | 0.25 | 0.02 | 0.01 | 0.02 | 0.01 | 0.02 | 0.08 | 0.13 | 0.2 | 0.23 |
|  | $Q_{obs}$ | 50% | 0.25 | 0.25 | 0.38 | 0.65 | 0.13 | 0.04 | 0.02 | 0.02 | 0.03 | 0.14 | 0.22 | 0.26 | 0.18 |
|  |  | 80% | 0.16 | 0.21 | 0.28 | 0.29 | 0.04 | 0.01 | 0.01 | 0.01 | 0.004 | 0.09 | 0.14 | 0.19 | 0.02 |
|  | $GW_{sim}$ | 50% | 21.6 | 20.6 | 20 | 20 | 20.3 | 20.7 | 21 | 21.3 | 21.7 | 21.8 | 21.6 | 21.5 | 20.99 |
|  |  | 80% | 18.7 | 16.5 | 14.8 | 15.6 | 16.2 | 16.8 | 17 | 17.9 | 18.3 | 17.7 | 17.7 | 18.1 | 18.08 |
|  | $GW_{obs}$ | 50% | 21.8 | 21.2 | 20.6 | 19.9 | 20.2 | 21.3 | 22.1 | 22.8 | 22.9 | 22.6 | 22.5 | 22.1 | 21.7 |
|  |  | 80% | 20 | 19.7 | 19.1 | 18.5 | 18.3 | 19.2 | 20 | 20.7 | 21.2 | 21.1 | 21.1 | 20.4 | 19.8 |
| Kiakola | $Q_{sim}$ | 50% | 0.39 | 0.59 | 0.67 | 0.17 | 0.05 | 0.03 | 0.03 | 0.23 | 0.11 | 0.18 | 0.24 | 0.29 | 0.25 |
|  |  | 80% | 0.24 | 0.45 | 0.53 | 0.05 | 0.004 | 0.01 | 0.01 | 0.02 | 0.04 | 0.09 | 0.11 | 0.15 | 0.04 |
|  | $Q_{obs}$ | 50% | 0.35 | 0.53 | 0.63 | 0.13 | 0.03 | 0.1 | 0.02 | 0.1 | 0.14 | 0.19 | 0.24 | 0.32 | 0.22 |
|  |  | 80% | 0.21 | 0.35 | 0.42 | 0.04 | 0.003 | 0.03 | 0.01 | 0.01 | 0.07 | 0.12 | 0.14 | 0.19 | 0.03 |

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
