# Peer review of "Quantifying Positive and Negative Human-Modified Droughts in the Anthropocene: Illustration with Two Iranian Catchments"

_water, doi:10.3390/w11050884_

Round 1

Reviewer 1 Report

The manuscript deals with the support water management to reduce drought impacts by further quantification of drought in the human influenced era through evaluation of positive and negative human activates on the hydrologic system. The manuscript presents important topic, but the other hand before taking decision about acceptance it should be corrected according to the following remarks. It should be mentioned in the title, that it concerns two Iranian catchments, so it is a form of case of study. State clearly how You get the data used in the study. The article lack a discussion on method, it should be mentioned what has been done earlier, also what approach and the other types of methods, that could have been used. On what bas You chose the used test? Lack of discussion of other data that could have been included or why they are excluded. Add references to all of the formula, in the case You are not the Author of the formula. Many of the headings are repeated in the second section, it should be rearranged and need to be rephrased. The axis in all the figures should be precisely described, unit, etc. The quality of the figures should be improved, the font size is too small, all figures should be similar to each other. Authors present a lot of figures and results, but at the same time there is very little discussion of the presented results. Authors should enhance the motivation, focus, innovation, introduction and conclusions of the work before to resubmit again the paper.

Why Authors don’t mention in the references and in the state-of art their own article concerning the presented issue published in 2018?

The article concern the same catchments of Kiakola and Eskandari,

What is the difference between this two methods, as it is quite similar to each other,

Positive and negative human-modified droughts: a quantitative approach illustrated with two Iranian catchments Elham Kakaei1,3, Hamid Reza Moradi1, Ali Reza Moghaddam Nia2, Henny A.J. Van Lanen3

Hydrol. Earth Syst. Sci. Discuss., https://doi.org/10.5194/hess-2018-124 Manuscript under review for journal Hydrol. Earth Syst. Sci. Discussion started: 29 May 2018 c

Author(s) 2018. CC BY 4.0 License.

Clarify the novelty, if exist, in an appropriate way, with comparison to the aforementioned published article.

Author Response

Response to Reviewer 1 Comments

 Point 1: The manuscript deals with the support water management to reduce drought impacts by further quantification of drought in the human influenced era through evaluation of positive and negative human activates on the hydrologic system. The manuscript presents important topic, but the other hand before taking decision about acceptance it should be corrected according to the following remarks.

Response 1: We appreciate very much the reviewer’s view that the paper addresses an important topic for water management, i.e. positive and negative human-modified droughts.

 Point 2: It should be mentioned in the title, that it concerns two Iranian catchments, so it is a form of case of study

Response 2: We have changed the title: “Quantifying negative and positive human-modified droughts in the Anthropocene: illustration with two Iranian catchments” (Lines 2-4). We would like to stress that the paper is merely a paper on an innovative extension of a methodology rather than a case study for two Iranian catchment. The latter are used to illustrate the methodology.

Point 3: State clearly how you get the data used in the study.

Response 3: The type and source of the data used (observed and modelled data) were described in section 3.1 (Page 6, L194- Page 7, L213).

Point 4: The article lack a discussion on method, it should be mentioned what has been done earlier, also what approach and the other types of methods that could have been used. On what base you chose the used test? Lack of discussion of other data that could have been included or why they are excluded.

 Response 4:

The existing approaches were described in introduction section and the downsides of each approach were explained (Page 2, L72-Page 3, L110). In addition, the example of application of approaches were described in (Page 21, L510- L570).

The extension of the human-modified drought methodology by Van Loon and Van Lanen (2013; 2015) and Van Loon et al. (2016), is innovative. Hence, it is hard to compare with earlier papers using this method, to compare with other methods, or to use data from other catchments. On the other hands, there is not a completely similar research to this article. But, the proposed methodology by Van Loon and Van Lanen (2013) have been applied all around the world and described in discussion section (Page 21, L510- Page 22, L551).

In addition, the various challenging issues of application of current mythology described (Section 5; Page 21, L522-L545).

Point 5: Add references to all of the formula, in the case you are not the Author of the formula. Many of the headings are repeated in the second section, it should be rearranged and need to be rephrased.

 Response 5:

The Eq. 1 was parented by Tallaksen et al. (2009) (Page 9, L309). Eq. 2 - 4 in formula format referring to Step 4, are not provided elsewhere, but are based upon existing knowledge (Van Loon and Van Lanen, 2013; 2015; Van Loon et al., 2016). We have referred to these in the revised manuscript (Page 9, L322).

The Method section was rewritten completely to eliminate the repeated headlines.

Point 6: The axis in all the figures should be precisely described, unit, etc.

Response 6:

The axis were revised, completely.

Point 7: The quality of the figures should be improved, the font size is too small, and all figures should be similar to each other.

Response 7:

We have improved the quality of the figures, by increasing the font. In addition, Figures 8-10 were located in pages with landscape orientation to be more visible. Because the aforementioned figures contain a large number of information and should not be compressed.

Point 8: Authors present a lot of figures and results, but at the same time there is very little discussion of the presented results.

 Response 8:

In current research was performed for a long-time period on daily basis. There were too many examples of detected drought types for both catchments. But, we presented only one example for each type and each catchment (Figures 8-10) and tried to show several events in each figure. We revised the result and discussion sections and discussed the results to some extent (Sections 4.3 (Page 15, L410- Page 19, L472& Section 5 (Page 20, L477- Page 21, L509)). However, we would like to keep the focus on the methodology rather to go in depth on the two Iranian catchment.

Point 9: Authors should enhance the motivation, focus, innovation, introduction and conclusions of the work before to resubmit again the paper.

 Response 9:

The manuscript revised and rewritten completely to enhance the motivation, focus, innovation, introduction and conclusions of the work.

Point 10: Why Authors don't mention in the references and in the state-of art their own article concerning the presented issue published in 2018? The article concern the same catchments of Kiakola and Eskandan, What is the difference between this two methods, as it is quite similar to each other, Positive and negative human-modified droughts: a quantitative approach illustrated with two Iranian catchments Elham Kakaeil ,3, Hamid Reza Moradil, Ali Reza Moghaddam Nia2, Henny A.J. Van Lanen3 Hydro!. Earth Syst Sci. Discuss., https://doi.org/10.5194/hess-2018-124 Manuscript under review for journal Hydro!. Earth Syst. Soi. Discussion started: 29 May 2018 c Author{s) 2018 CC BY 4.0 License. https://www.hydror-earth-syst-sci-discuss.net/hess-2018-124/hess-2018-124 pdf Clarify the novelty, if exist, in an appropriate way, with comparison to the aforementioned published article.

 Response 10:

The manuscript Kakaei et al. (2018) has been submitted to HESSD. But the manuscript has been withdrawn by the authors and Copernicus Publishers removed it. Please check the below link of HESS Journal:

https://www.hydrol-earth-syst-sci-discuss.net/hess-2018-124/

You you will read “This preprint has been withdrawn”.

References

Tallaksen, L.M.; Hisdal, H.; Van Lanen, H.A.J. Space–time modelling of catchment scale drought characteristics. Journal of Hydrology. 2009, 375, 363-372, doi: 10.1016/j.jhydrol.2009.06.032

Van Loon, A.F.; Van Lanen, H.A.J. Making the distinction between water scarcity and drought using an observation-modeling framework. Water Resour. Res. 2013, 49, 1483–1502, doi: 10.1002/wrcr.20147.

Van Loon, A.F.; Van Lanen, H.A.J. Testing the observation-modelling framework to distinguish between hydrological drought and water scarcity in case studies around Europe. European Water. 2015, 49: 65-75.

Van Loon, A. F.; Gleeson, T.; Clark, J.; Van Dijk, A. I. J. M.; Stahl, K.; Hannaford, J.; Di Baldassarre, G.; Teuling, A. J.; Tallaksen, L.M.; Uijlenhoet, R.; Hannah, D.M.; Sheffield, J.; Svoboda, M.; Verbeiren, B.; Wagener, T.; Rangecroft, S.; Wanders, N.; Van Lanen, H.A.J. Drought in the Anthropocene. Nature Geosci. 2016, 9, 89–91, doi:

Reviewer 2 Report

The manuscript entitled "Quantifying negative and positive human-modified droughts in Anthropocene: a case study" presents the method of distinguishing different types of drought, with the two catchments in Iran as study areas. I am really impressed by the authors' huge effort in conducting this study. Therefore, I would recommend this manuscript to be accepted for publication. But, prior to its acceptance, there are a couple of issues should be addressed. Detail comments are stated below:

In the "abstract" section, please add a few sentences to state the gap in knowledge that is going to be addressed in this study. This can highlight the significance of this study.

In the "introduction" section, the authors should explain why it is necessary to propose a new method to distinguish different types of drought. Are there any existing methods to distinguish different types of drought in academia? If yes, what are they? Also, what are the limitations of those methods? The authors should address the above questions as readers may wonder why this study should be conducted. Besides, the term "the positive effects of human-modified drought" is a bit confusing to me. It takes me some time to know what it means. Indeed, the term is rarely used in academia. Why not using a more reader-friendly term, say drought mitigation, to talk about it? 

I suggest Section 2 to be "study area," while Section 3 is "materials and methods," and then other sections are renumbered accordingly. This is not good to merge "materials and methods" and "study area" together into the same section, as it may distort the flow of idea in the manuscript. Besides, in the "study area" section, lines 130-131: "the emphasis in this study is on the methodology, and the results from the catchment are just to explain the methodology." I disagree with these sentences. This is because readers may wonder whether the proposed method only works well in a specific area. If the authors want to emphasize the merit of their proposed method, it is necessary to highlight the representativeness of their study area (catchments), as well as whether other regions in the world have a similar setting to their study area. Or else, readers may think the proposed method is only applicable in the two catchments (Kiakola and Eskandari).

In the "discussion" section, I suggest the authors compare their findings with those in other studies (say, those studies using other methods in distinguishing different types of drought). On the one hand, this can help authors to verify their findings by cross-comparison. On the other hand, this can further show the merits of their proposed methods (if the proposed method performs better). I can see that the authors have attempted to do this (lines 560-575). But, there is much room for improvement. 

I am not a native English speaker. But, I find several typos and grammatical mistakes in the manuscript, which may affect the readability of the manuscript. I recommend the authors to have the manuscript proofread by a professional editor prior to its submission.

Author Response

Response to Reviewer 2 Comments

Point 1: The manuscript entitled "Quantifying negative and positive human-modified droughts in Anthropocene: a case study" presents the method of distinguishing different types of drought, with the two catchments in Iran as study areas. I am really impressed by the authors' huge effort in conducting this study. Therefore, I would recommend this manuscript to be accepted for publication. But, prior to its acceptance, there are a couple of issues should be addressed.

Response 1: We appreciate very much the reviewer’s view. The manuscript revised according to the recommendations.

Point 2: In the "abstract" section, please add a few sentences to state the gap in knowledge that is going to be addressed in this study. This can highlight the significance of this study.

Response 2: The abstract was rewritten and the aim of research were explained (L16-L21). In addition, the introduction was rewritten completely, and the gaps in knowledge and aims of application of the extended methodology were described (Page 2, L46-L57; Page 3, L99-L110).

Point 3: In the "introduction" section, the authors should explain why it is necessary to propose a new method to distinguish different types of drought. Are there any existing methods to distinguish different types of drought in academia? If yes, what are they? Also, what are the limitations of those methods? The authors should address the above questions as readers may wonder why this study should be conducted. Besides, the term "the positive effects of human-modified drought" is a bit confusing to me. It takes me some time to know what it means. Indeed, the term is rarely used in academia. Why not using a more reader-friendly term, say drought mitigation, to talk about it?

Response 3: the introduction was rewritten completely to address the importance of developing new method (Page 1, L40- Page 2, L57) and describing the existing method (Page 2, L72- Page 3, L98).

The term of Positive and negative human-modified drought were introduced in current research. The positive and negative words were used to express the kind of human impact activities on the system, while these interventions are intended only for the purpose of exploitation, not for improving the status of the system. In addition, the authors used "positive human-modified" phrase instead of drought mitigation to make it clear and emphasis on the main aim of the current research, which is quantifying positive and negative human modified drought in Anthropocene.

Point 4: I suggest Section 2 to be "study area," while Section 3 is "materials and methods," and then other sections are renumbered accordingly. This is not good to merge "materials and methods" and "study area" together into the same section, as it may distort the flow of idea in the manuscript. Besides, in the "study area" section, lines 130-131: "the emphasis in this study is on the methodology, and the results from the catchment are just to explain the methodology." I disagree with these sentences. This is because readers may wonder whether the proposed method only works well in a specific area. If the authors want to emphasize the merit of their proposed method, it is necessary to highlight the representativeness of their study area (catchments), as well as whether other regions in the world have a similar setting to their study area. Or else, readers may think the proposed method is only applicable in the two catchments (Kiakola and Eskandari).

 Response 4:

The study areas moved to the section 2.

As mentioned in manuscript (Page 3, L103-L110), the expanded method was applied to Iranian catchments that climate-induced droughts are common and usually have to cope with human interventions that cause the drought characteristics to change, either in a positive or in a negative way (Page 3, L117-L119). Despite the focus of current research was on Iran, the methodology is not unique to Iran and can be applied in other areas around the world (Page 23, L605-606). Also, examples of the application of "observed-modelled" approach all around the word have presented in discussion section (Page 21, L510- Page 22, L551). In addition, in study area and discussion sections, complete explanation was given to determine why these two regions were selected and what kind of the human-interventions affect the hydrological system (Page 3, L127-L140; Page 4, L150-L166; Page 22, L571-L596).

Point 5: In the "discussion" section, I suggest the authors compare their findings with those in other studies (say, those studies using other methods in distinguishing different types of drought). On the one hand, this can help authors to verify their findings by cross-comparison. On the other hand, this can further show the merits of their proposed methods (if the proposed method performs better). I can see that the authors have attempted to do this (lines 560-575). But, there is much room for improvement

Response 5:

The discussion part was rewritten. The extension of the human-modified drought methodology by Van Loon and Van Lanen (2013; 2015) and Van Loon et al. (2016), is innovative. Hence, it is hard to compare with earlier papers using this method, to compare with other methods, or to use data from other catchments. On the other hands, there is not a completely similar research to this article. But, the proposed methodology by Van Loon and Van Lanen (2013) have been applied all around the world and described in discussion section (Page 21, L510- Page 22, L551). In addition, the example of application of other approaches were described in (Page 21, L552- L570),

Point 6: I am not a native English speaker. But, I find several typos and grammatical mistakes in the manuscript, which may affect the readability of the manuscript. I recommend the authors to have the manuscript proofread by a professional editor prior to its submission.

 Response 6:

The manuscript was revised and rewritten completely to improve the English mistakes.

References

Van Loon, A.F.; Van Lanen, H.A.J. Making the distinction between water scarcity and drought using an observation-modeling framework. Water Resour. Res. 2013, 49, 1483–1502, doi: 10.1002/wrcr.20147.

Van Loon, A.F.; Van Lanen, H.A.J. Testing the observation-modelling framework to distinguish between hydrological drought and water scarcity in case studies around Europe. European Water. 2015, 49: 65-75.

Van Loon, A. F.; Gleeson, T.; Clark, J.; Van Dijk, A. I. J. M.; Stahl, K.; Hannaford, J.; Di Baldassarre, G.; Teuling, A. J.; Tallaksen, L.M.; Uijlenhoet, R.; Hannah, D.M.; Sheffield, J.; Svoboda, M.; Verbeiren, B.; Wagener, T.; Rangecroft, S.; Wanders, N.; Van Lanen, H.A.J. Drought in the Anthropocene. Nature Geosci. 2016, 9, 89–91

Round 2

Reviewer 1 Report

my numerous remarks were included

Reviewer 2 Report

All of my comments in the previous report have been addressed. I recommend this manuscript to be accepted for publication.

Water EISSN 2073-4441 Published by MDPI AG, Basel, Switzerland RSS E-Mail Table of Contents Alert
Back to Top